# Women Walk in High Heels: Lumbar Curvature, Dynamic Motion Stimuli and Attractiveness

**DOI:** 10.3390/ijerph18010299

**Published:** 2021-01-03

**Authors:** Norbert Meskó, Fanni Őry, Edit Csányi, Lea Juhász, Gréta Szilágyi, Olivér Lubics, Ádám Putz, András Láng

**Affiliations:** 1Institute of Psychology, University of Pécs, 7624 Pécs, Hungary; oryfanni@gmail.com (F.Ő.); csanyi.edit2@gmail.com (E.C.); juhaszlea19980816@gmail.com (L.J.); putz.adam@pte.hu (Á.P.); lang.andras@pte.hu (A.L.); 2Institute of Psychology, Károli Gáspár University of the Reformed Church in Hungary, 1037 Budapest, Hungary; szilagyigreti97@gmail.com; 3Institute of Psychology, Eötvös Loránd University, 1075 Budapest, Hungary; lubics.oliver@gmail.com

**Keywords:** lumbar curvature, high-heeled shoes, attractiveness, mate choice preference, dynamic/video stimuli

## Abstract

Previous studies have demonstrated that the angle of women’s lumbar curvature affects men’s attractiveness judgments of them. The theoretically optimal angle of lumbar curvature provides better resistance against both hyperlordosis and hypolordosis as biomechanical costs of a bipedal fetal load that could impair a woman’s fertility. Since men find this attribute attractive, women aim to emphasize it by wearing high-heeled shoes. The primary objective of the present study was to test this evolutionary hypothesis using short videos presenting women walking by the camera. In line with previous findings based on static stimuli (photographs), dynamic stimuli (videos) presenting women walking in high-heeled shoes were expected to elicit increased attractiveness ratings as compared to women wearing flat shoes, which would be associated with the angle of lumbar curvature. Videos were taken of 52 female models walking in two conditions (i.e., wearing either high-heeled or flat shoes). A total of 108 participants (61 males, 47 females) rated the walking models’ physical attractiveness in an online setting. Each model’s lumbar curvature was measured both in high heels and in flat shoes using photographs taken of them prior to each video recording. The results showed that wearing high heels consistently increased the models’ attractiveness, regardless of whether or not it decreased their natural difference from the theoretically optimal angle of lumbar curvature. Both male and female observers showed this positive effect. Furthermore, a negative correlation was found between the models’ body mass index (BMI) and their perceived attractiveness scores in both conditions.

## 1. Theoretical Background

From an adaptationist view, it is a consistent finding across various cultures that men’s judgments of their potential female partners are primarily based on the targets’ physical condition in connection with bearers’ biological qualities (e.g., health, fecundity) [1,2]. According to a broader evolutionary logic, universal beauty standards of the female body are determined by a set of anthropomorphic cues including the body mass index (BMI), the waist-to-hip ratio (WHR), the shoulder-to-waist ratio (SWR), the waist-to-stature ratio (WSR), and the size of the breasts and the buttocks [3]. The related studies have shown that these phenotypic cues are the most reliable indicators of a woman’s fertility, health, youth, and probable parental investment [4].

Wearing high-heeled shoes is a common practice among women in the Western world, despite the fact that it involves various health risks [5,6,7]. Although Dilley, Hockey, Robinson, and Sherlock [8] clearly demonstrated peer and workplace pressure on women to wear high-heeled shoes, a study conducted in the United Kingdom found that 67% of the female participants regularly wore high heels at work by their own choice, whereas only 13% of them reported that they merely complied with a requirement or dress code by doing so [9]. At the same time, Barnish, Morgan, and Barnish [10] found in a meta-analysis that wearing high heels might provide certain psychosexual benefits for women. Maarouf [11] also reported that Egyptian women consistently perceived high-heeled shoes as a factor increasing a woman’s attractiveness, regardless of the observers’ job or career stage. In a real-life experiment, male passers-by responded kindlier to a young female stranger who wore high heels compared to one wearing flat shoes, and male but not female participants were also more likely to help the female confederate in the high-heel condition [12]. Guéguen, Stefan and Renault [13] found that both male and female participants in a forced-choice evaluation of women wearing high vs. flat heels selected the former as the more attractive, the sexier, the younger, the prettier, and the more elegant (even though the shoes and feet were not visible in the pictures). These findings support the hypothesis that high-heeled shoes increase their wearer’s perceived attractiveness by emphasizing the secondary sexual characteristics of the female body.

Most recently, Prokop [14] found in line with previous studies that both men and women assigned higher attractiveness ratings to women wearing high-heeled shoes. Participants also perceived heeled legs as longer than non-heeled ones. The author concluded that visually increased leg length might promote intrasexual rivalry between women for access to the opposite sex through the enhancement of sexual attractiveness. Reynolds et al. [9] found that 98% of the 932 female participants reported willingness to wear high heels at parties or clubs. Women’s preference for wearing high-heeled shoes may be explained by its potential to enhance the wearer’s attractive physical features by changing her body posture.

Lewis et al. [15] were the first to demonstrate that high-heeled shoes change the angle of the wearer’s lumbar curvature, that is, the wedging in the third-to-last lumbar vertebra. This area of the lower back is responsible for absorbing and transmitting the weight of the body [16]. With age, the spinal column undergoes significant physical changes. Amonoo-Kuofi [17] points out that the curvature of women’s lumbar spine gradually and moderately decreases from the late twenties until the thirties, when an increase begins, and a second decrease takes place from the fifties on. Due to its physiological function, the lumbar curvature also adapts to the weight it has to support. According to Okanishi, Kito, Akiyama, and Yamamoto [18], an increase in body mass induces postural changes affecting the relative position of the head, the neck and the trunk, which compensate for the shift in the center of gravity. A similar adjustment occurs during pregnancy when the altered mass distribution causes an anterior pelvic tilt. In terms of human evolution, it was crucial for ancient women to survive these changes during pregnancy and deliver healthy offspring.

An increased wedging of the lumbar vertebrae extends the tailbone relative to the rest of the spine, which results in a more acute angle of lumbar curvature [19]. Lewis et al. [19] argue that men’s evolutionary preference for women with a lumbar curvature between the angles of hypo-, and hyperlordosis provided them with a reproductive advantage, since these women were less likely to suffer from spinal injuries while being more likely to sustain multiple pregnancies without serious complications. The orthopedic literature suggests that the ideal angle of lumbar curvature is approximately 45° [20]. Based on related findings, Lewis and colleagues [19] manipulated the degree of female models’ lumbar wedging by creating five morphs representing the naturally occurring range of lumbar curvature in the population (varying between 14° and 69°). The authors found that men assigned the highest attractiveness ratings to those female models whose vertebral wedging was closest to the optimum. These results suggest that men indeed show evolutionary preference for an optimal lumbar curvature in women.

Similar to virtual photo editing tools, high-heeled shoes may also be used to change the degree of the vertebral wedging. To test this potential of high heels, Lewis et al. [15] extended the above research using profile images of women in heels and flats. The authors argued that women would use high-heeled shoes as a means to increase their attractiveness by manipulating their lumbar curvature. The results confirmed the authors’ hypothesis: high-heeled shoes as compared to flat shoes increased both women’s lumbar curvature and their physical attractiveness judged by men. However, the researchers discovered a lumbar-curvature-dependent effect on perceived attractiveness. Namely, wearing high heels increased attractiveness only when it reduced (rather than increased) the difference between a woman’s natural lumbar curvature and the optimal angle.

A more recent study published by Pazhoohi, Doyle, Macedo, and Arantes [21] revealed that both male and female observers were sensitive to changes in female models’ lumbar curvature, although men were significantly more responsive to a more pronounced vertebral wedging, which supports the hypothesis that an arching back serves as a signal of sexual proceptivity in women. Furthermore, male participants found pictures of women taken in a side or rear-side view to be more attractive than their front-side counterparts.

One’s perceived physical attractiveness is not solely determined by independent traits (such as the shape or weight of the body, proportions of the body parts, etc.) but also by the complex interaction of these features (for a review, see Swami and Furnham, [22,23]). Furthermore, the human body very rarely appears as a static stimulus in the social environment, but it is usually seen in motion [24,25]. Thus, an important factor in terms of perceived attractiveness is the internal consistency of the motion and the coherence between individual features [26]. Internal consistency is the degree of synchronization between an individual’s anthropometric and kinematic parameters. A relatively high level of internal consistency significantly increases a model’s attractiveness, since it conveys honest signals of maturity, health, and reproductive fitness [27].

If lumbar curvature is indeed an important adaptive signal of the attractiveness of the female body, then dynamic stimuli (e.g., video recordings of walking women) should elicit responses similar to those associated with static stimuli (e.g., pictures of women). The aim of the present study was not merely the replication of the findings of Lewis et al. [15] but to methodologically extend it by using short videos presenting walking women instead of photographs. With this, dynamic stimulus materials were used for the first time to our knowledge to study the effect of lumbar curvature on perceived attractiveness.

## 2. Hypotheses

### 2.1. Hypothesis 1

The results of Lewis et al. [15] showed that young women’s lumbar curvature was higher when wearing high-heeled compared to flat-soled shoes. The same finding was expected in the present study, that is, the female models’ lumbar curvature was predicted to be higher when wearing high heels vs. flats as measured in the pictures taken of each model.

### 2.2. Hypothesis 2

Numerous studies demonstrated the positive effect of wearing high-heeled shoes on women’s perceived attractiveness [12,13,14,15]. In line with these findings, participants of the present study were expected to judge a female model as more attractive when she walked by the camera in high heels than when she walked in flats. Although no previous study used dynamic stimuli to the authors’ knowledge, Lewis et al. [15] demonstrated the expected effect with pairs of photographs of women. While the effects of motion, including dance and walking, on the perception of the female body have been thoroughly investigated [26,27,28,29], the effects of changes in lumbar curvature caused by wearing high heels while walking have never been the main focus of research. The present study was aimed at replicating the findings of Lewis et al. [15] by using short videos presenting walking women instead of photographs.

### 2.3. Hypothesis 3

According to the evolutionary hypothesis, the optimal angle of a woman’s lumbar curvature is an adaptive signal, which has had an essential impact on men’s mate choice preferences [15,19]. In the present study, however, women were also expected to be sensitive to this signal, considering that assessing a potential same-sex rival’s reproductive value has adaptive benefits [3,21,30]. At the same time, it is likely that women do not perceive the sexual attraction of other women as a mere threat. We need to consider that women may also be able to assess the physical beauty of other women in a similar way to men because of their more fluid sexuality (e.g., bisexual attraction). Thus, no significant difference was expected between male and female observers’ attractiveness ratings.

### 2.4. Hypothesis 4

In line with the findings reported by Lewis and colleagues [15,19], Hypothesis 4 predicted that wearing high heels would only increase those female models’ perceived attractiveness whose natural difference from the optimal angle of vertebral wedging (i.e., approximately 45.5°) would decrease as a result.

## 3. Method

The preparation of the study strictly followed the methodological steps applied by Lewis et al. [15] (see Study 2). The only exception was that two short videos of each model while walking in heels/flats were recorded to be used as dynamic stimuli, in addition to the two photographs taken of each.

The research protocol was licensed by the Hungarian United Ethical Review Committee (under License No. 2019/91). In line with the Declaration of Helsinki, all participants gave their informed consent for inclusion before they participated in the study. The models signed a declaration of consent, while their raters gave consent online, prior to evaluating the models.

### 3.1. Stimulus Materials

A total of 52 female university students aged 18 to 24 years (*M* = 19.98, *SD* = 1.28) were recruited as models from the first year of the psychology BA program at the University of Pécs, Hungary. As compensation, each student earned extra course credits. The models’ major biometrical data were as follows. Their height ranged from 154 to 182 cm (*M* = 167.47, *SD* = 5.75); their weight ranged from 47 to 93 kg (*M* = 61.17, *SD* = 10.29); the BMI ranged from 16.2 to 31.8 kg/m^2^ (*M* = 21.84, *SD* = 3.62). BMI was independent of lumbar curvature both in flats (*r* = −0.09, *p* = 0.577) and in high heels (*r* = 0.04, *p* = 0.817).

At the time of preparing the stimulus materials, none of the models had or expected a baby. One model reported to have spondylolisthesis, two suffered from scoliosis, and one revealed that she had undergone back surgery before. In addition to these four models, four further models were excluded from the sample because their clothing did not comply with the standards of the study.

Following the procedure employed by Lewis and colleagues [15], two outliers were excluded from further data analysis (see for details section Data preparation), whose wearing of heels resulted in an increase in lumbar curvature greater than 2.4° standard deviations, that is, 9.144° (*M* = 1.15°, *SD* = 3.81°). It has not yet been clarified how wearing high heels may cause such an atypical change in lumbar curvature, but it is possibly due to a lack of experience in wearing heels [15]. One model was excluded from the sample due to her high-heeled shoes failing to meet the minimum required heel height, as a result of which they did not have an impact on the model’s lumbar curvature. The final stimulus set included the videos and pictures of 42 models.

### 3.2. Procedure

The videos and pictures of the models were prepared on a previously agreed date and time at an adequately equipped laboratory. The models were instructed to wear a tight-fitting unicolor dark T-shirt and either slim-fit jeans or leggings. Furthermore, they were asked to bring their own high-heeled shoes measuring 8 to 10 cm from the sole to the tip of the heel. To eliminate possible effects of hairstyle [31,32], each model tied up her hair with a hairband prior to the photoshoot and filming.

### 3.3. Photographs and Video Recordings

A photograph and a video of each model were prepared in each of two conditions, that is, when wearing flat shoes and when wearing high-heeled shoes. The photograph was taken first in each condition. The models were instructed to stand with their right shoulder facing the wall during the photoshoot. During the video recording, they kept the same orientation and walked by the camera across a 5 m long passage at a slow, comfortable and even pace. In line with Lewis et al. [15], the body portions above the shoulders and below mid-calf were removed from each photograph and video using the Pixelmator Photo software. This procedure served to protect the models’ privacy, and to eliminate the possible unwanted effects of the size of the heels and the increase in height and relative leg length when wearing heels as compared to flats (see Appendix A).

### 3.4. Participants and Attractiveness Assessment

A total of 108 participants aged 18 to 43 years (*M* = 23.73, *SD* = 4.5) rated the attractiveness of the models presented in the short videos. The sample included 61 males (56.5%) and 47 females (43.5%). By relationship status, 42 participants (38.9%) were single at the time of data collection, 6 (5.6%) had casual encounters but no permanent relationship, 25 (23.1%) had a long-term relationship without cohabitation, 30 (27.8%) cohabited with their partners, and 5 (4.6%) were married. By level of education, 87 participants (80.6%) reported currently attending tertiary education, while 21 (19.4%) did not have a student status. Tertiary education was completed by 37 participants (34.26%) and secondary education by 71 (65.74%).

Each participant was presented with the overall stimulus set, including 84 videos, in an online setting. The order of the stimuli was randomized for each participant with the PsyToolkit software [33,34]. The participants rated each model’s attractiveness in each condition (heels/flats) using a 10-point scale ranging from extremely unattractive (1) to extremely attractive (10).

### 3.5. Lumbar Curvature Measurement

The models’ lumbar curvature was measured with a virtual protractor tool (Screen Protractor, Iconico, Inc., Phoenix, AZ, US), following the protocol employed by Lewis et al. [15], who adopted it from clinical orthopedic practice [35]. Lumbar curvature was measured as the angle determined by two separate lines drawn on the side-view pictures of each model, the first line placed parallel to the top of the lower back and the second line parallel to the top of the buttocks.

## 4. Results

### 4.1. Data Preparation

The variable in focus was not the angle of vertebral wedging per se, but the change in the difference between each model’s observed lumbar curvature and the optimal value of 45.5° caused by wearing high heels as compared to flats [19]. The relevant measure was obtained by first calculating the absolute difference between 45.5° and each model’s lumbar curvature observed when (1) wearing flat-soled shoes and when (2) wearing high-heeled shoes, and then subtracting (2) from (1). A positive value indicated that the model’s lumbar curvature was closer to the optimum in heels than in flats (*N* = 33, *M* = 2.49°, *SD* = 2.02°), whereas a negative value indicated that wearing high-heeled shoes increased the difference between the observed lumbar curvature and the optimal value (*N* = 9, *M* = −3.7°, *SD* = 2.05°). The difference between the two groups was tested with an independent-samples *t*-test (*M*_Diff_ = 6.20, *SE*_Diff_ = 0.76, *t*(40) = 8.14, *p* < 0.001, *d* = 2.73).

The impact of heels on attractiveness was obtained for each model by subtracting the mean perceived attractiveness measured in the flats condition from that measured in the heels condition. Thus, positive and negative values indicated higher perceived attractiveness in the heels condition and in the flats condition, respectively. The above measures obtained for the 42 models included in the final stimulus set were used to test the hypotheses of the study.

Sample size calculation showed that, in order to achieve the statistical power of 0.80 at the significance level of α = 0.05 for an effect size of d = 0.50 (i.e., the smallest effect size for attractiveness ratings in Lewis et. al, 2017), 34 models would have been needed. Thus, the total sample of 42 models was deemed as adequate to detect the expected magnitude of effect with higher statistical power or to detect a smaller effect size with the expected power.

### 4.2. Hypothesis Testing

According to result of a paired samples t-test, in line with Hypothesis 1, the models showed significantly greater lumbar curvature in heels (*M* = 36.27°, *SD* = 8.26°) than in flats (*M* = 35.10°, *SD* = 8.21°, *t*(41) = 2.326, *p* = 0.025, *d* = 0.14). Hypotheses 2 and 3 were tested with a 2 × 2 repeated measures ANOVA with the raters’ gender (male vs. female) and the models’ footwear (heels vs. flats) as within-subject factors. The results revealed no significant interaction effect (*F*(1,41) = 0.002, *p* = 0.967, partial *η^2^* < 0.001) neither a significant main effect for raters’ gender (*F*(1,41) = 1.014, *p* = 0.320, partial *η^2^* = 0.024). The analysis revealed a significant main effect of footwear on perceived attractiveness (*F*(1,41) = 14.134, *p* = 0.007, partial *η^2^* = 0.163). Irrespective of raters’ gender, higher attractiveness ratings were assigned to models in the high heels condition (*M* = 5.09, *SD* = 1.33) than in the flats condition (*M* = 4.92, *SD* = 1.31, *d* = 0.13). In other words, male and female raters equally perceived the walking models’ lumbar curvature as a more salient attractiveness signal when increased by high heels than when not enhanced artificially.

In order to test Hypothesis 4, which predicted that wearing high heels would only increase perceived attractiveness if reducing the target’s natural difference from the optimal angle of lumbar curvature, the models were sorted into two groups according to whether or not wearing high heels decreased the difference in focus (positive values indicating a decrease, while zero and negative values indicating a lack thereof), and the predicted difference in perceived attractiveness between the two groups was tested with an independent-samples *t*-test. Contrary to the prediction, the results revealed no significant difference in perceived attractiveness between those whose natural difference from the optimum was reduced (*M* = 4.66, *SD* = 1.37) vs. not reduced (*M* = 5.21, *SD* = 1.32, *t*(40) = 1.09, *p* = 0.281). That is, wearing heels consistently increased the models’ perceived attractiveness, regardless of whether or not their lumbar curvature was closer to the optimum in the heels condition as compared to the flats condition.

## 5. Discussion

To the authors’ knowledge, the present study is the first to use video stimuli to test the impact of wearing high heels on female targets’ perceived attractiveness. The related results demonstrated that the walking female models presented in side view were perceived as more attractive when wearing heels than when wearing flats, while the footwear was not visible in the videos (H1). That is, the positive association between wearing heels and perceived attractiveness previously demonstrated with static images also holds true when the female body is observed in motion. This finding is important particularly because the human body most frequently appears in the social environment as a dynamic rather than static stimulus [24]. Although similar findings were obtained with dynamic stimuli in a previous study [30], the authors used point-light stimuli presenting walking women’s biological motion (assessed with biomechanical analysis) and not female bodies as seen under natural conditions.

As predicted, no gender difference was found between observers in the positive effect of high heels on perceived attractiveness (H2), which is in line with previous findings [21,30]. This supports the idea that adequately perceiving the sexually attractive features of the female body is an essential part of women’s mate choice strategies and intrasexual rivalry [36]. In these terms, wearing high-heeled shoes may be considered as a means of female intrasexual rivalry aimed at gaining access to high-quality partners [37,38,39]. As an alternative explanation, however, we should note that there may be other reasons for the similar response pattern of women and men in our research. According to the theory of female sexual fluidity [40], women may find their same-sex peers sexually attractive simply because, under certain circumstances, they may appear as potential partners in the social space.

The prediction that wearing heels as compared to flats would increase young women’s lumbar curvature was confirmed by the related results (H3). This finding is in line with that obtained with static images by Lewis and colleagues [15], who concluded that the observed increase in the female models’ perceived attractiveness was due to the increase in the angle of lumbar curvature. The validity of this conclusion is further supported by the stimuli employed in the present study, since the raters could not see the models’ footwear in the videos, therefore the increased attractiveness ratings in the heels condition may only be explained by the change in the models’ posture, that is, in their lumbar curvature.

Contrary to the previous finding that wearing high heels only increased perceived attractiveness in cases when it decreased the target’s natural difference from the optimal angle of lumbar curvature (i.e., 45.5°; [15]), the present study revealed that wearing heels consistently increased the models’ attractiveness, regardless of its impact on the difference in question (H4). A possible explanation for this discrepancy lies in the methodological differences between the two studies. Namely, while the measures of lumbar curvature were based on photographs in both studies, the attractiveness ratings in the present study were obtained in response to video stimuli. It is possible that the videos were not fully consistent with the respective photographs in terms of the observable angles of lumbar curvature, albeit presenting the same models in the same footwear. Furthermore, it may be much more difficult to adequately process the lumbar curvature of a moving body presented in a short video than when exploring a static image. The potential methodological difficulties are well illustrated by an eye-tracking study reported by Pazhoohi, Doyle, Macedo, and Arantes [21], who had to control the computer-generated images for the model’s BMI score and waist-to-hip ratio in order to eliminate their effects on the observers’ responses to the systematic manipulation of lumbar curvature. It is also possible that the number of models (*N* = 9) for whom the lumbar curvature further departed from the optimal in high heels was too small to detect even a medium size effect. Therefore, this result of the study might be treated with caution and Hypothesis 4 should be tested again in further studies.

## 6. Limitations and Future Directions

An important assumption underlying the employed methodology was that the angles of lumbar curvature measured in the photographs were consistent with the respective angles observable in the video recordings. There is no evidence, however, that this was indeed the case. Nor did any of the related previous studies report reliable consistency between the employed static and dynamic stimuli. This is an important limitation of the study, which may be overcome in future studies by obtaining attractiveness ratings for both types of matched stimuli and assessing the correlation between the two measures.

Another important limitation is posed by the narrow age range of the models. Following Lewis and colleagues [15], the present study only involved young female models, whereas many older women wear heels as a matter of course. Considering the natural age-related decrease in lumbar curvature beginning around the early fifties, future studies should examine the impact of wearing high heels on the lumbar curvature and perceived attractiveness of women aged 40 to 50 years. A particularly interesting question to be clarified by this line of research is whether and how these measures are associated with the individual variability in women’s reproductive potential.

The results of the present study are consistent with the mating interest hypothesis assigning adaptive value to men’s responsiveness to observable cues of women’s sexual exploitability [41] but not with the pregnancy hypothesis attributing the attractiveness of the optimal angle of lumbar curvature to its adaptive benefit in terms of childbearing [15]. Specifically, wearing heels consistently increased the models’ perceived attractiveness, regardless of whether or not it decreased their natural difference from the optimal angle of lumbar curvature. It has yet to be clarified why this previously demonstrated “optimum effect” was not elicited by the walking female models involved in the present study. Further studies are needed to systematically explore the impact of lumbar curvature on the perceived attractiveness of the walking female body.

## 7. Conclusions

This study demonstrates how an evolutionarily anchored hypothesis [19] and its empirical investigation [15] may be adapted for a more ecologically valid setting by using short videos presenting the biological motion of female bodies. The obtained results corroborate some of the related previous findings, while raising further questions concerning the relationship between the static and dynamic components of the impact of young women’s lumbar curvature on their perceived attractiveness.

## Data Availability

The data that support the findings of this study are available from the corresponding author, upon reasonable request.

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
