# Peer review of "Women Walk in High Heels: Lumbar Curvature, Dynamic Motion Stimuli and Attractiveness"

_ijerph, 2021, doi:10.3390/ijerph18010299_

Round 1
Reviewer 1 Report
I am satisfied with the authors' corrections to their manuscript.
Author Response
Thank you for your review
Reviewer 2 Report
Although the authors revised the manuscript, there are some points the author did not address accordingly:
- No tables or figures in the results, only the text showing the results would be challenging for the audience;
- It would be great if the author can use a figure to explain the measurement of the lumbar curvature angle in the methods part;
- The author did not pay the effort to explain the evidence and the mechanisms of how the high heels shoes would change the lumbar curvature angle in the introduction part;
- The author did not explain how to normalize different high heels shoes, flat shoes to the findings in the methods part.
Since there are a lot of concerns did not resolve in the manuscript and I will not recommend it for publication.
Author Response
we list our concerns with regard to the comments of Reviewer 2 point by point.
- No tables or figures in the results, only the text showing the results would be challenging for the audience.
- With regard to results, the scientific rule is either to present them in text or in tables/figures to avoid redundancy. We deemed the applied statistical test to be simple and commonly used tests, thus, results should be easily understood by those who regularly read scientific articles.
- It would be great if the author can use a figure to explain the measurement of the lumbar curvature angle in the methods part.
- Measurement of the lumbar curvature is explained in lines 210-214. We believe that this description is detailed enough for the audience to follow the line of reasoning. We fully agree with the reviewer that a more detailed explanation could be needed for replication. For this purpose audience is trusted to the work of Lewis et al. (2017) as indicated in line 211.
- The author did not pay the effort to explain the evidence and the mechanisms of how the high heels shoes would change the lumbar curvature angle in the introduction part.
- We believe that lines 78-89 explain both the anatomical mechanism and the evolutionary importance of change in lumbar curvature.
- The author did not explain how to normalize different high heels shoes, flat shoes to the findings in the methods part.
- The same issue was raised by the reviewer previously. Our response (copy-pasted): “The brand and type of shoes were not standardized. But each participant was asked to bring their own high-heeled shoes with a heel size determined in advance. The flat shoes were really flat, with no heels. The heels of the high heels were 8-10 cm high in all cases.” Since change in lumbar curvature was measured and because body portions above the shoulders and below mid-calf were removed, we deemed it unnecessary to include this description in the manuscript.
Reviewer 3 Report
Thanks authors, the considerations made have been included.
Author Response
Thank you for your review
This manuscript is a resubmission of an earlier submission. The following is a list of the peer review reports and author responses from that submission.
Round 1
Reviewer 1 Report
I enjoyed and appreciate how much empirical evidence was cited in the introduction. The study has a strong rationale and the method is solid. I have a few concerns about the hypotheses, analyses, and data interpretation. But these should each be addressable.
Please see my specific comments below:
Abstract – “...harmful environmental factors that could impair a woman’ fertility” Give some examples. It’s unclear what you mean by this statement alone.
p. 1 – the first paragraph of “Theoretical background” appears to be template text.
p. 1 – “The global beauty standards of the female body...” This makes it sound like beauty is the only thing that matters, when what is meant is that physical attractiveness tends to be prioritized by men over other traits. It may be helpful to describe how this reference supports the authors’ statement.
p.2 “These findings [support] the hypothesis...” It’s more difficult than this to confirm a hypothesis. Support is a better word here.
p. 3 – Emphasize at the end of the introduction that use of dynamic stimuli is a unique contribution to this literature. You not only replicate Lewis et al.; you expand upon it.
Though I’m sympathetic to the hypothesis that “both men and women will perceive women in high heels as more attractive than those in flat shoes, but for different reasons”, I’m a bit disappointed that the study design did not include supplementary tests of whether men’s and women’s responses were the same 1) because they both perceived the women as more attractive, vs. 2) men found them more attractive and women found them threatening. Bisexuality and sexually fluid attraction is non-negligible among women. So, I don’t think the authors can conclude that women found women in high heels more attractive BECAUSE of intrasexual competition unless the alternative explanation is ruled out. Consider rephrasing this throughout (i.e., in the introduction and discussion) and/or acknowledge this limitation in the discussion.
I don’t entirely understand why hypothesis 5 is needed. Are the authors testing whether lumbar curvature explains attractiveness above and beyond BMI? This seems like a far more interesting question to test than simply seeing whether the BMI and lumbar curvature are related. Given that the authors report an association between the two, it would be interesting to see whether variance in lumbar curvature uniquely explains attractiveness ratings after controlling for BMI.
p. 5 – The authors need to provide a power analysis showing that their sample size is sufficient for their planned analyses.
p. 5 – How long did it take participants to complete the full set of ratings? 84 videos could be rather fatiguing.
p. 6 – please provide the exact p value for each statistical test.
p. 6 – Why were t-tests used to assess hypotheses 1 + 2 when the main effects of the 2x2 design would provide this information? Reporting both is redundant, and ANOVA stats are better to report because they allow for better comparison of the interaction with main effects.
p. 6 – provide a measure of effect size for the ANOVA findings (e.g., partial eta-squared).
p. 6 – It’s worth noting that for Hypothesis 4 analyses the two groups were Ns = 33 and 9. The second group is fairly small (and likely wouldn’t give the analyses sufficient power). So, the null effect reported for this hypothesis has good potential to be a Type II error. This should be mentioned in the discussion.
Reviewer 2 Report
This study investigated women walk in high heels shoes and lumbar curvature via the photo or dynamic motion video to identify the attractiveness. A lot of statements are based on the choice of male selection/preference aspect, this will arise the concern for the gender equality argument among the public, especially for the feminist audience. I don’t believe some statements in the manuscript will be acceptable by the audience. Additionally, no figures or tables were shown for the results/findings in this manuscript. Since a lot of concerns shown up in this manuscript, I will not recommend publication in the current format. Below are the questions and concerns:
Introduction
Lines 96-97, the author mentioned wearing high heels shoes by the lumbar curvature angle, while little evidence and the mechanisms of how the high heels shoes would change the lumbar curvature angle were shown in the introduction part.
Hypothesis 1
Lines 123 – 127, since the previous research have reported young women’s lumbar curvature was higher when wearing high-heeled compared to flat-soled shoes. What is the point of conducting the same comparison protocol in this study? Any novel findings differ from previous studies?
Method
Lines 188-189, did the high heels shoes and flat shoes tested in the study were in the same type, height, brand and model? If not, how did you normalize different high heels shoes, flat shoes to the findings?
Lines 215 – 219, please specify how you measure the lumbar curvature angle in more detail. Did you measure it during the static standing or during the walking?
Results
No tables or figures were found in this section. Only the text statement is not sufficient.
Reviewer 3 Report
I appreciate the opportunity to review this article, so it can be a contribution to knowledge on this topic, I value its implementation but I consider it to be a new version of Lewis et al. (2017) Why Women Wear High Heels: Evolution, Lumbar Curvature, and Attractiveness.
The research includes videos instead of photos. I think it would be very appropriate to expand and emphasize all the differences, and the new that is provided in the article, so that it is not a simple replica of the research mentioned above, the only difference being the use of video stimuli to have real evidence of biological movement of the bodies.
It would be interesting to expand by highlighting these differences at least in the conclusions, starting from line 343.